# A Newcastle disease virus expressing a stabilized spike protein of SARS-CoV-2 induces protective immune responses

Weina Sun [1], Yonghong Liu[1], Fatima Amanat [1,2], Irene González-Domínguez [1], Stephen McCroskery[1,3], Stefan Slamanig [1], Lynda Coughlan [4,5], Victoria Rosado[1], Nicholas Lemus[1], Sonia Jangra[1,6], Raveen Rathnasinghe[1,2,6], Michael Schotsaert [1,6], Jose L. Martinez[1], Kaori Sano[1], Ignacio Mena [1,6], Bruce L. Innis [7], Ponthip Wirachwong[8], Duong Huu Thai[9], Ricardo Das Neves Oliveira[10], Rami Scharf[7], Richard Hjorth[7], Rama Raghunandan[7], Florian Krammer [1,11], Adolfo García-Sastre [1,3,6,11,12] & Peter Palese [1,3 ✉]

Rapid development of COVID-19 vaccines has helped mitigating SARS-CoV-2 spread, but more equitable allocation of vaccines is necessary to limit the global impact of the COVID-19 pandemic and the emergence of additional variants of concern. We have developed a COVID-19 vaccine candidate based on Newcastle disease virus (NDV) that can be manufactured at high yields in embryonated eggs. Here, we show that the NDV vector expressing an optimized spike antigen (NDV-HXP-S) is a versatile vaccine inducing protective antibody responses. NDV-HXP-S can be administered intramuscularly as inactivated vaccine or intranasally as live vaccine. We show that NDV-HXP-S GMP-produced in Vietnam, Thailand and Brazil is effective in the hamster model. Furthermore, we show that intramuscular vaccination with NDV-HXP-S reduces replication of tested variants of concerns in mice. The immunity conferred by NDV-HXP-S effectively counteracts SARS-CoV-2 infection in mice and hamsters.

[1] Department of Microbiology, Icahn School of Medicine at Mount Sinai, New York, NY 10029, USA. [2] Graduate School of Biomedical Sciences, Icahn School of Medicine at Mount Sinai, New York, NY 10029, USA. [3] Department of Medicine, Icahn School of Medicine at Mount Sinai, New York, NY 10029, USA. [4] University of Maryland School of Medicine, Department of Microbiology and Immunology, Baltimore, MD 21201, USA. [5] University of Maryland School of Medicine, Center for Vaccine Development and Global Health (CVD), Baltimore, MD 21201, USA. [6] Global Health Emerging Pathogens Institute, Icahn School of Medicine at Mount Sinai, New York, NY 10029, USA. [7] PATH, Center for Vaccine Access and Innovation, Washington, DC 20001, USA. [8] The Government Pharmaceutical Organization, Bangkok 10400, Thailand. [9] Institute of Vaccines and Medical Biologicals, Nha Trang City, Khanh Hoa Province, Vietnam. [10] Instituto Butantan, São Paulo, SP 05503-900, Brazil. [11] Department of Pathology, Icahn School of Medicine at Mount Sinai, New York, NY 10029, USA. [12] The Tisch Cancer Institute, Icahn School of Medicine at Mount Sinai, New York, NY 10029, USA. ✉email: peter.palese@mssm.edu

The coronavirus disease 2019 (COVID-19) pandemic caused by severe acute respiratory syndrome coronavirus 2 (SARS-CoV-2) has brought disastrous outcomes to public health, education, and economics worldwide. Emerged variants of concern that are currently circulating could threaten the prior preventive achievements if not managed properly. The rollout of COVID-19 vaccines such as mRNA vaccines (Pfizer and Moderna), inactivated virus vaccines (Sinovac, Sinopharm), adenovirus-vector vaccines (AstraZeneca, CanSino Biologics, Gamaleya Research Institute, and J&J) have helped to contain the spread of the virus tremendously, stressing the importance of prophylactic measures. However, despite the high efficacy of mRNA vaccines, the availability of such vaccines to developing countries is restricted owing to cold or ultra-cold chain requirements and a lack of manufacturing infrastructure and capacity. Indeed, with North America and Europe having the highest vaccination rates, vaccine resources are much less accessible to developing countries in Latin America, Asia, and Africa[1]. Such inequitable availability of vaccine delays prompt control of COVID-19 and increases the risk of additional variants to emerge. This highlights the urgent need for affordable vaccines that can be produced locally.

We have previously developed a Newcastle disease virus (NDV)-based COVID-19 vaccine, in which a membrane-anchored spike protein is expressed on the surface of the NDV virion. This NDV vector could be used either as a live vaccine or an inactivated vaccine[2,3]. Here, we describe a next-generation version of the NDV vector expressing a prefusion spike protein stabilized by Hexa Pro (HXP) mutations, which are reported to contribute to high protein yield, favorable confirmation, and enhanced stability[4]. This construct is designated NDV-HXP-S. As an egg-based vaccine like the influenza virus vaccine, NDV-HXP-S is suitable for large-scale production to cover a fair share of global demands. A survey conducted by The World Health Organization (WHO) estimated the production capacity for pandemic influenza vaccines (monovalent) could reach ~4.15 billion doses in 12 months by 31 established manufacturers worldwide, among which 28 manufacturers have egg-based facilities producing 79% of total doses[5]. This report realistically reflects the feasibility of manufacturing a large quantity of NDV-based COVID-19 vaccine since few modifications to the influenza virus vaccine manufacturing process are needed. Moreover, this estimation assumes the dose of the NDV-based vaccine required will match that of monovalent pandemic influenza vaccines (15 µg/0.5 mL), without adjusting for the potential antigen-sparing effect of adjuvants[5]. A safe and inexpensive adjuvant could likely expand the number of doses per egg.

In the belief that NDV-HXP-S could be the solution for self-sufficient supplies of COVID-19 vaccine in many low- and middle-income countries, in this study, we thoroughly evaluated the immunogenicity and protective efficacy of live and inactivated NDV-HXP-S in preclinical mouse and Golden Syrian hamster models. We assessed the beta-propiolactone (BPL)-inactivated NDV-HXP-S whole-virion vaccine (GMP manufactured in Thailand, Vietnam, and Brazil) via the intramuscular (IM) route using a two-dose regimen. In addition, we explored the possibility of using the NDV-HXP-S as a live vaccine either via the intranasal (IN) route or a combination of IN and IM routes. These preclinical studies showed that the NDV-HXP-S vaccine in its live or inactivated format was highly immunogenic, inducing potent binding and neutralizing antibodies (NAbs), which offered protection against SARS-CoV-2 replication or SARS-CoV-2-induced disease in vivo. The high levels of NAbs induced by NDV-HXP-S allowed the variants of concern/interest (B.1.1.7, B.1.351, or P.1) to be neutralized, despite the fact that a reduction of neutralization titer was observed against the B.1.351 variant. A similar level of reduction has been shown in other vaccine cohorts as well[6–9]. Importantly, Good Manufacturing Practice (GMP)-grade NDV-HXP-S vaccine lots produced by manufacturers from Vietnam (Institute of Vaccines and Medical Biologicals, IVAC), Thailand (Government Pharmaceutical Organization, GPO), and Brazil (Instituto Butantan) showed excellent immunogenicity and protective efficacy in the hamster model, demonstrating the consistency of the manufacturing process in different locations and the possibility of mass production. Vaccine trials with the inactivated vaccines have been started in Thailand (NCT04764422, HXP-GPOVac) and Vietnam (NCT04830800, COVIVAC) and the live vaccine is currently in clinical development in Mexico (NCT04871737, Patria).

## Results

**NDV expressing a membrane-bound prefusion stabilized spike protein**. To ensure high immunogenicity of the spike antigen expressed by the NDV vector, we improved the spike construct by introducing the prefusion stabilizing Hexa Pro (HXP) mutations that were identified and characterized in an earlier study[4]. Specifically, we added the HXP mutations into our previously described S-F chimera (HXP-S), in which the polybasic cleavage site was removed and the transmembrane domain and cytoplasmic tail of the spike were replaced with those from the fusion (F) protein of NDV. The nucleotide sequence of the construct was codon-optimized for mammalian host expression. The HXP-S sequence was inserted between the P and M genes of the NDV genome and the virus was rescued (Fig. 1a). We concentrated the virus in the allantoic fluid through a sucrose cushion and performed a sodium dodecyl–sulfate polyacrylamide gel electrophoresis (SDS–PAGE) to visualize NDV viral proteins as well as the presence of the spike protein by Coomassie Blue staining. Compared with the WT NDV, the NDV-HXP-S showed an extra band between 160 kD and 260 kD below the L protein of the NDV that corresponds to the size of the uncleaved S0 (Fig. 1b). To further demonstrate that the spike protein is associated with NDV virions, we subsequently fractionated the concentrated virus by ultracentrifugation through a continuous sucrose gradient. The rationale is that if the spike protein is separated from the NDV particles, it would be in different fractions from those containing the NDV proteins. After ultracentrifugation, a total of 26 fractions were collected from the top (10% sucrose) to the bottom (60% sucrose), with 1 mL per fraction, was collected and resolved on SDS–PAGE followed by Coomassie Blue staining. In fact, we observed that the spike protein co-migrated with NDV viral proteins, confirming that the spike is indeed incorporated into NDV virions (Supplementary Fig. 4).

**Vaccination with inactivated NDV-HXP-S via the intramuscular route induces protective immune responses in mice and hamsters**. To evaluate the immunogenicity and protective efficacy of the inactivated NDV-HXP-S, we first performed a dose-ranging study in mice that were "sensitized" by IN administration of a non-replicating human adenovirus 5 expressing human angiotensin-converting enzyme 2 (Ad5-hACE2). Mice are not naturally susceptible to SARS-CoV-2 infection, but gene delivery of hACE2 to the lungs of mice using a viral vector such as Ad5-hACE2, can sensitize mice to subsequent infection with SARS-CoV-2[10].

BPL-inactivated NDV-HXP-S with or without an adjuvant CpG 1018[11] was tested in the study. Specifically, we immunized BALB/c mice with low doses of NDV-HXP-S at a total protein content of 1 µg, 0.3 µg, 0.1 µg, 0.03 µg, and 0.01 µg per mouse without the adjuvant. In the adjuvanted groups, mice were vaccinated with either 0.1 µg or 0.03 µg of the vaccine in

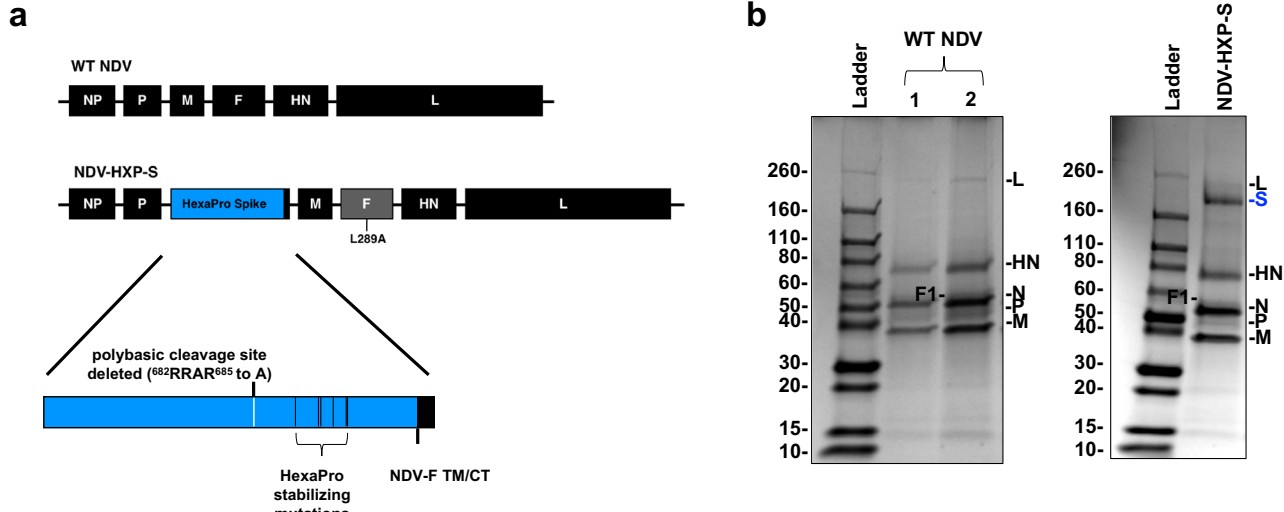

**Fig. 1 Design of the NDV-HXP-S construct. a** Structure and design of the NDV-HXP-S genome. The ectodomain of the spike was connected to the transmembrane domain and cytoplasmic tail (TM/CT) of the F protein (blue: the ectodomain of the spike; black: NDV components; gray: NDV F gene with L289A mutation). The original polybasic cleavage site was removed by mutating RRAR to A. The HexaPro (F817P, A892P, A899P, A942P, K986P, and V987P) stabilizing mutations were introduced. The sequence was codon-optimized for mammalian host expression. **b** Protein staining of NDV-HXP-S. WT NDV as well as NDV-HXP-S were partially purified from the allantoic fluid through a sucrose cushion and resuspended in PBS. In all, 5 μg (1) and 10 μg (2) of the WT NDV, as well as 10 μg of NDV-HXP-S were resolved on 4–20% SDS–PAGE. The viral proteins were visualized by Coomassie Blue staining (L, S[Blue], HN, N, P, and M). A representative image out of more than three independent experiments is shown.

combination with 10 μg or 30 μg of CpG 1018 per mouse. Mice that were immunized with 1 μg of the inactivated WT NDV (vector-only) were used as negative controls. The inactivated vaccine was administered via the intramuscular route following a prime-boost regimen at a 3-week interval (Fig. 2a). Mice were bled to measure spike-specific serum IgG. The prime immunization showed a dose-dependent antibody response, in which the adjuvanted groups appeared to develop higher antibody responses than those which received the same amount of vaccine without the adjuvant. The antibody titers were greatly enhanced after the boost in all the animals that were vaccinated with NDV-HXP-S, resulting in no significant differences among the groups (with a marginal dose-dependent trend). Of note, mice that were vaccinated with as low as 0.03 μg and 0.01 μg of vaccine per animal also developed good IgG titers, suggesting the vaccine is highly immunogenic (Fig. 2b). Further evaluation of IgG2a over IgG1 ratio in post-boost sera from three selected groups, group 1 (1 μg) representing animals that received the non-adjuvanted vaccine, group 7 (0.1 μg + 30 μg CpG 1018) representing animals that received the adjuvanted vaccine and group 10 (WT NDV) as the negative controls, suggested a $T_H1$-biased immune response[12]. The CpG 1018 appeared to reinforce a $T_H1$-biased immune response, given that a more-pronounced difference between the IgG2a and IgG1 levels was observed in the adjuvanted group as compared to that in the non-adjuvanted group (Supplementary Fig. 1a). To examine the neutralizing activity of the immune sera from mice, pooled sera from group 1, 7, and 10 were tested. Neutralization titers against the prototype (WT) USA-WA1/2020 virus as well as two SARS-CoV-2 variants, B.1.1.7 and B.1.351 were measured (Fig. 2c). We observed potent neutralizing activity of mouse sera to the WT strain from group 1 ($ID_{50} = 669$) and group 7 ($ID_{50} = 627$). As expected, mouse sera from both groups neutralized the B.1.1.7 variant equally well (group 1 $ID_{50} = 652$; group 7 $ID_{50} = 914$). An approximately fivefold reduction of neutralization titers was observed toward the B.1.351 variant (group 1 $ID_{50} = 132$; group 7 $ID_{50} = 132$). Our result is in agreement with decreased neutralization titers of

mRNA vaccines that have been reported[6–9], as the E484K mutation in the B.1.351 variant is mainly responsible for the resistance to NAbs[13,14]. To assess protection conferred by vaccination, mice were treated with Ad5-hACE2[10]. Five days after the Ad5-hACE2 treatment, mice were challenged with $10^5$ plaque-forming unit (PFU) of USA-WA1/2020 SARS-CoV-2. Infectious viral titers in the lungs of challenged animals at day 2 and day 5 post challenge were measured as the readout of protection. We observed a significant drop of the viral titer in the lung homogenates of all vaccinated animals compared to those from the negative control group, also in a dose-dependent manner. On day 5 post infection, infectious viruses were cleared in all NDV-HXP-S-vaccinated animals (Fig. 2d). These results support the conclusion that the inactivated NDV-HXP-S induced strongly protective antibody responses directed to the spike protein of SARS-CoV-2, with CpG 1018 as a possible dose-sparing adjuvant.

In the hamster model, we evaluated two doses of inactivated NDV-HXP-S without adjuvants, with CpG 1018 or AddaVax as the adjuvant. Each hamster was vaccinated intramuscularly with a total of 5 μg of NDV-HXP-S. The negative control group was vaccinated with 5 μg of inactivated WT NDV. The healthy control group was kept unvaccinated. After two doses with a 3-week interval, hamsters were challenged with $10^4$ PFU of the USA-WA1/2020 strain, except that the healthy control group was mock-challenged (Fig. 3a). Change of body weight, viral titers in the lung homogenates, and nasal washes at day 2 and day 5 post challenge were measured to evaluate protection. In addition, spike-specific IgG titers in the post-prime and post-boost sera, as well as neutralizing activity in the post-boost sera were measured by enzyme-linked immunosorbent assay (ELISA) and microneutralization assay, respectively. NDV-HXP-S was found to be highly immunogenic in hamsters as well, with both CpG 1018 and AddaVax increasing serum IgG (Fig. 3b). Without any adjuvant, NDV-HXP-S vaccinated hamsters developed potent NAbs in the post-boost sera to the prototype (WT) SARS-CoV-2 ($ID_{50} = 2429$) and the B.1.1.7 variant ($ID_{50} = 2710$). AddaVax

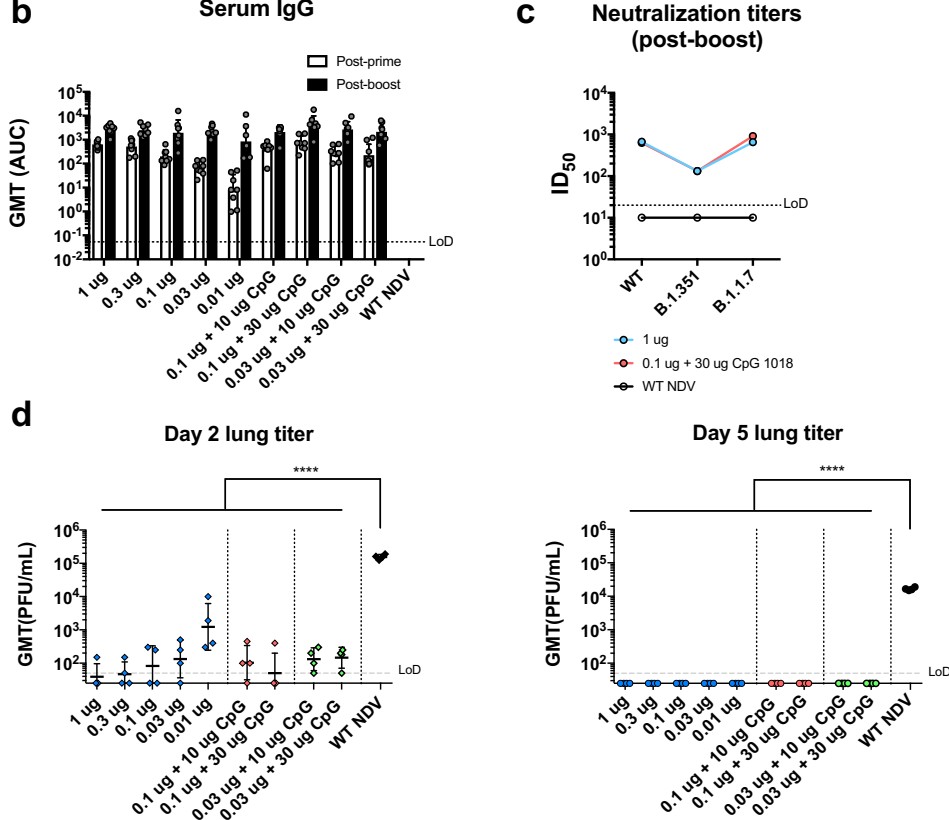

**Fig. 2 Low doses of inactivated NDV-HXP-S induce a protective antibody response in mice. a** Design of the study. Nine-to-ten-week-old female BALB/c mice were used. Group 1–5 ($n = 8$) were vaccinated with unadjuvanted NDV-HXP-S at 1 μg, 0.3 μg, 0.1 μg, 0.03 μg, and 0.01 μg per mouse, respectively. Group 6 and 7 ($n = 8$) were vaccinated with 0.1 μg of NDV-HXP-S with 10 or 30 μg of CpG 1018 per mouse, respectively. Group 8 and 9 ($n = 8$) were vaccinated with 0.03 μg of NDV-HXP-S with 10 or 30 μg of CpG 1018 per mouse, respectively. Group 10 ($n = 8$) was vaccinated with 1 μg of WT NDV as the negative control. The vaccine was administered via the intramuscular (I.M.) route at D0 and D21. Blood was collected at D21 and D43. Mice were sensitized with Ad5-hACE2 at D45 and challenged with $10^5$ PFU of the USA-WA1/2020 strain. **b** Spike-specific serum IgG. Antibodies in post-prime (D21) and post-boost (D43) sera were measured by ELISAs. Geometric mean titer (GMT) represented by the area under the curve (AUC) was graphed, with geometric standard deviation (SD) as error bars. **c** Neutralizing activity of serum antibodies. Post-boost sera from group 1, group 7, and group 10 were pooled within each group and tested in neutralization assays against the USA-WA1/2020 strain (WT), the B.1.351 variant, and B.1.1.7 variant in technical duplicate. Serum dilutions inhibiting 50% of the infection ($ID_{50}$) were plotted. (LoD: limit of detection; LoD = 1:20; An $ID_{50}$ = 1:10 was assigned to negative samples). **d** Viral load in the lungs. Lungs of a subset of animals ($n = 4$) from each group were collected on day 2 and day 5 post challenge. The whole lungs were homogenized in 1 mL of PBS. Viral titers were measured by plaque assay on Vero E6 cells and plotted as GMT of PFU/mL (LoD = 50 PFU/mL; a titer of 25 PFU/mL was assigned to negative samples). The error bars represent geometric SD. Statistical difference was analyzed by ordinary one-way ANOVA corrected for Dunnett's multiple comparisons test (****$p < 0.0001$). (blue: unadjuvanted groups; red: 0.1 μg adjuvanted with CpG 1018; green: 0.03 μg adjuvanted with CpG 1018).

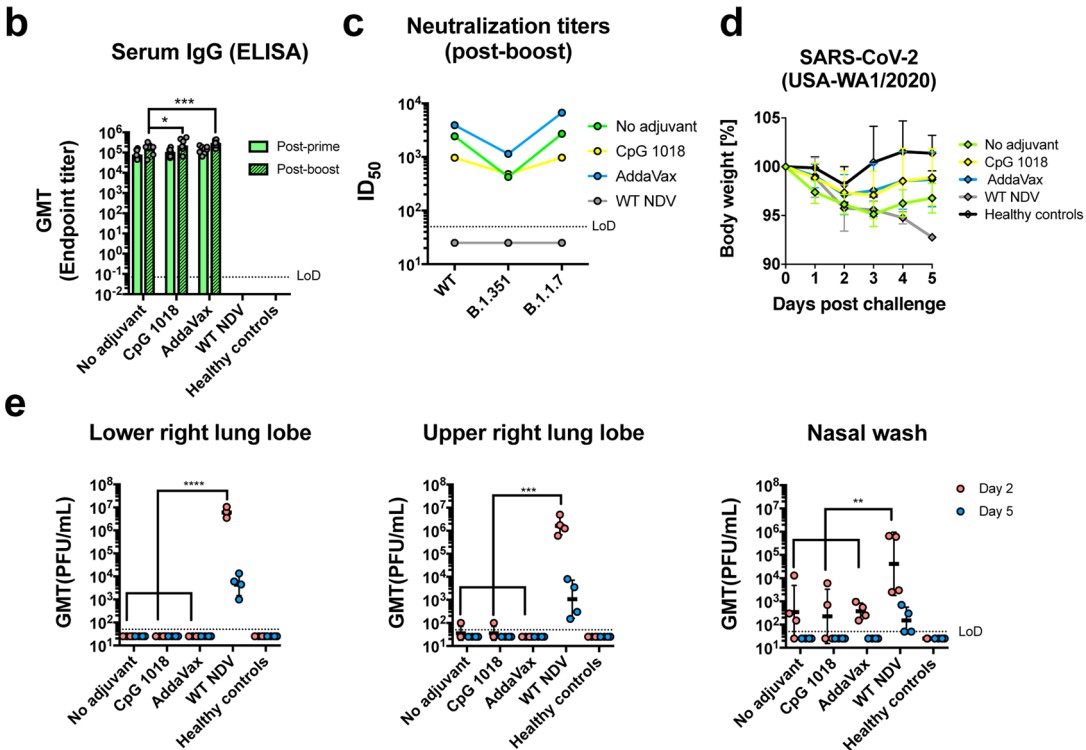

**Fig. 3 Inactivated NDV-HXP-S induces protective antibody response in hamsters. a** Design of the study. Eighteen-to-twenty-week-old female Golden Syrian hamsters were used. Groups 1–3 ($n = 8$) were vaccinated with 5 μg of NDV-HXP-S without adjuvants, with CpG 1018 and AddaVax, respectively. Group 4 ($n = 8$) was vaccinated with 5 μg of WT NDV as the negative control. Group 5 ($n = 6$) was not vaccinated. The vaccine was administered via the intramuscular (I.M.) route at D0 and D21. Blood was collected at D21 and D39. Group 1–4 were challenged with $10^4$ PFU of USA-WA1/2020 strain at D42. Group 5 was mock-challenged with PBS. **b** Spike-specific serum IgG. Antibodies in post-prime (D21) and post-boost (D39) sera were measured by ELISAs. GMT endpoint titer was graphed. The error bars represent geometric SD. Statistical difference was analyzed by two-way ANOVA corrected for Dunnett's multiple comparisons test (*$p = 0.0262$; ***$p = 0.0006$). **c** Neutralizing activity of serum antibodies. Post-boost sera from groups 1–4 were pooled within each group and tested in neutralization assays against USA-WA/2020 strain (WT), B.1.351 variant, and B 1.1.7 variant in technical duplicate (green: no adjuvant; yellow: adjuvanted with CpG 1018; blue: adjuvanted with AddaVax; gray: WT NDV control). Serum dilutions inhibiting 50% of the infection (ID$_{50}$) were plotted. (LoD = 1:50; an ID$_{50}$ = 1:25 was assigned to negative samples) **d** Body weight change of hamsters. Body weights were recorded for 5 days after challenge (green: no adjuvant; yellow: adjuvanted with CpG 1018; blue: adjuvanted with AddaVax; gray: WT NDV control; black: healthy controls). The error bars represent geometric SD. **e** Viral load in the lungs and nasal washes. The lower right and upper right lung lobes of a subset of animals ($n = 4$ for groups 1–4; $n = 3$ for group 5) from each group were collected at day 2 (red) and day 5 (blue) post challenge. Each lung lobe was homogenized in 1 mL PBS. Nasal washes were collected in 0.4 mL of PBS. Viral titers were measured by plaque assay on Vero E6 cells and plotted as GMT of PFU/mL (LoD = 50 PFU/mL; a titer of 25 PFU/mL was assigned to negative samples). The error bars represent geometric SD. Statistical difference was analyzed by two-way ANOVA corrected for Dunnett's multiple comparisons test (**$p = 0.0026$ (WT NDV vs. no adjuvant), $p = 0.0025$ (WT NDV vs. CpG 1018), $p = 0.0024$ (WT NDV vs. AddaVax); ***$p = 0.0001$; ****$p < 0.0001$).

enhanced NAb titers (ID$_{50}$ = 3913 to the WT; ID$_{50}$ = 6684 to the B.1.1.7 variant), but the CpG 1018 did not (ID$_{50}$ = 970 to the WT; ID$_{50}$ = 973 to the B.1.1.7 variant). Interestingly, the antibodies in the CpG 1018 group appeared to be more cross-neutralizing to B.1.351 with only a twofold decrease (ID$_{50}$ = 480). An approximately sixfold reduction of neutralizing titer to the B.1.351 (ID$_{50}$ = 425) was observed in the unadjuvanted group, whereas

approximately fourfold reduction of neutralizing activity to the B.1.351 (ID$_{50}$ = 1144) was observed in the AddaVax group (Fig. 3c). After the challenge, animals in both adjuvanted groups showed quicker recovery of the body weight than animals in the unadjuvanted group (Fig. 3d). Viruses were cleared in the lungs of all NDV-HXP-S vaccinated hamsters at day 2 except for one out of four animals in the unadjuvanted and CpG 1018 group, which

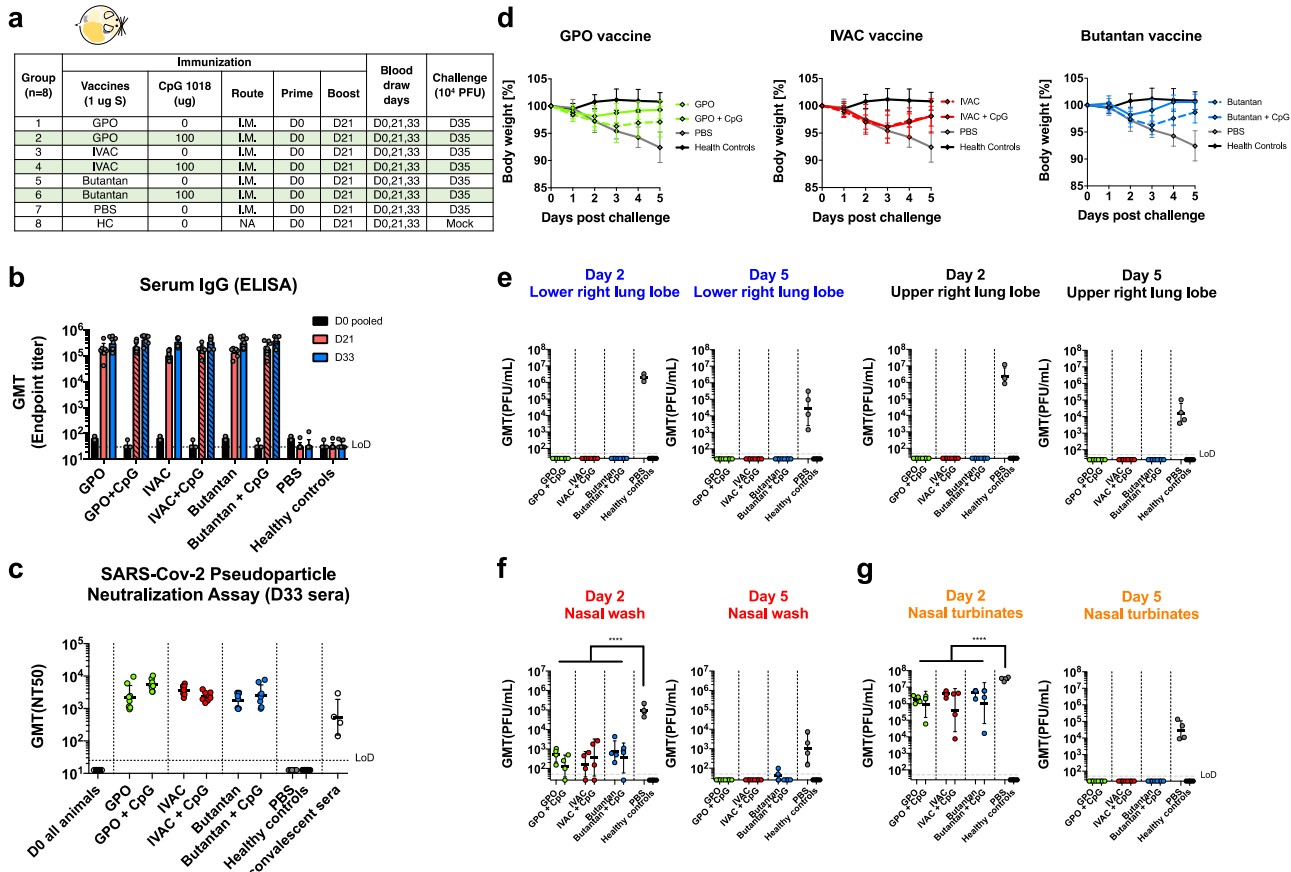

**Fig. 4 GMP lots of inactivated NDV-HXP-S produced by influenza virus vaccine manufacturers induce a protective antibody response in hamsters.**
**a** Design of the study. Nine-to-eleven-week-old female Golden Syrian hamsters were used. Groups 1–6 ($n = 8$) were vaccinated with 1 μg of spike antigen of inactivated NDV-HXP-S from GPO, IVAC, and Butantan in the absence or presence of CpG 1018. Group 7 ($n = 8$) was vaccinated with PBS as the negative control. Group 8 ($n = 8$) was not vaccinated (HC, healthy controls). The vaccine was administered via the intramuscular (I.M.) route at D0 and D21. Blood was collected at D0, D21, and D33. Groups 1–7 were challenged with $10^4$ PFU of the USA-WA1/2020 strain at D35. Group 8 was mock-challenged with PBS. **b** Spike-specific serum IgG. Antibodies in pre-vaccination (D0, black), post-prime (D21, red), and post-boost (D33, blue) sera were measured by ELISAs (solid bars: unadjuvanted; pattern bars: adjuvanted). GMT endpoint titers were graphed. The error bars represent geometric SD. **c** Neutralizing activity of serum antibodies. A pseudo-particle neutralization assay was performed by Nexelis to measure neutralization titers of post-boost sera (D33). Human convalescent sera were included in the same assay as the controls (LoD = 1:25; An $ID_{50}$ = 1:12.5 was assigned to negative samples). **d** Body weight changes of hamsters. Body weights were monitored for 5 days after the challenge. The error bars represent geometric SD. **e** Viral load in the lungs. The lower right and upper right Lung lobes of a subset of animals ($n = 4$) from each group were collected at day 2 and day 5 post challenge. Each lung lobe was homogenized in 1 mL PBS. **f** Viral load in nasal washes and **g** nasal turbinates. On day 2 and day 5 post challenge, nasal washes were collected in 0.4 mL of PBS. Nasal turbinates were homogenized in 0.5 mL PBS. Viral titers were measured by plaque assay on Vero E6 cells and plotted as GMT of PFU/mL (LoD = 50 PFU/mL; a titer of 25 PFU/mL was assigned to negative samples). The error bars represent geometric SD. Statistical difference was analyzed by ordinary one-way ANOVA corrected for Dunnett's multiple comparisons test (****$p < 0.0001$). **c–g** Green: GPO; red: IVAC; blue: Butantan; gray: PBS: black: healthy controls.

showed low titers that were very close to the limit of detection. The viral load became undetectable at day 5 post challenge in all NDV-HXP-S vaccinated hamsters. In addition, all NDV-HXP-S vaccinated groups showed a reduction of viral titers in the nasal washes compared with control animals (Fig. 3e). In conclusion, the inactivated NDV-HXP-S induced high levels of binding and neutralizing antibody responses in hamsters, significantly reduced viral titers in the lungs, and lowered virus shedding from the nasal cavity. Both CpG 1018 and AddaVax exhibited beneficial adjuvant effects.

**Formulations of inactivated NDV-HXP-S GMP-produced by GPO, IVAC, and Instituto Butantan are effective in a preclinical hamster study.** In the belief that the existing influenza virus vaccine manufacturers should be equipped to produce

NDV-HXP-S, we put this possibility to the test. In collaboration with three influenza virus vaccine manufacturers (IVAC, GPO, Instituto Butantan) from Vietnam, Thailand, and Brazil, pilot GMP lots of whole BPL-inactivated NDV-HXP-S vaccines were produced. We obtained these vaccine preparations and subsequently evaluated them in hamsters. Vaccines containing 1 μg of S antigen were administered intramuscularly to each hamster with or without 100 μg CpG 1018 as the adjuvant following a prime-boost regimen with a 3-week interval. A phosphate-buffered saline (PBS)-vaccinated negative control and a healthy control group were included. Two weeks after the boost, animals were challenged with $10^4$ PFU of the USA-WA1/2020 strain (Fig. 4a). Animals having received vaccines from different producers developed comparable binding IgG titers, whereas CpG 1018 showed marginal adjuvant effects (Fig. 4b). The neutralizing activity of serum antibodies after the boost was measured in a

pseudo-particle neutralization assay (PNA) at Nexelis (part of CEPI's global network of laboratories to centralize assessment COVID-19 vaccine candidates)[15]. Neutralization titers of hamster sera were substantially higher than those of human convalescent sera used as controls in the same assay (Fig. 4c). After the challenge, we observed that animals that were immunized with all three vaccines without the adjuvant developed a similar trend of body weight change. Animals that were immunized with GPO and Butantan vaccines in the presence of the CpG 1018 showed improved protection manifested by less weight loss. The adjuvant did not seem to alleviate body weight loss of animals that received the IVAC vaccine (Fig. 4d). This observation is consistent with the post-boost antibody level (D33) (Fig. 4b). Nevertheless, all the animals that were vaccinated with NDV-HXP-S developed immunity to inhibit SARS-CoV-2 replication in the lungs showing no detectable infectious viral titers (Fig. 4e). In the nasal washes of vaccinated animals, virus shedding was significantly reduced in contrast to that in the negative control animals (Fig. 4f). The virus was still able to replicate in the nasal turbinates of all infected animals, resulting in transient weight loss in the vaccinated animals. But all NDV-HXP-S-vaccinated animals showed a reduced viral load (Fig. 4g). To evaluate SARS-CoV-2-induced lung disease, the left lung lobes were collected at day 5 post challenge and processed for histopathology analysis. As expected, a much less-pathological change reflecting injury or inflammation was observed in the lungs of NDV-HXP-S vaccinated animals as compared with those in the lungs of negative control animals (Supplementary Fig. 2a–e). The SARS-CoV-2 antigen was also mostly cleared in vaccinated animals examined by immunohistochemistry staining for the N protein of SARS-CoV-2 (Supplementary Fig. 2f). In conclusion, inactivated NDV-HXP-S vaccine prepared by three egg-based influenza virus vaccine manufacturers showed equally good efficacy at inducing binding/NAbs, inhibiting virus replication and shedding as well as minimizing SARS-CoV-2-induced lung pathology in hamsters. An FDA-approved adjuvant, CpG 1018 mildly increased protection from bodyweight loss of hamsters after challenge. Regarding the lack of enhanced protection of CpG 1018 with IVAC vaccine compared to IVAC vaccine alone, we speculate that it might be a result of slightly different production methods of three manufacturers.

**Live NDV-HXP-S protects hamsters and mice from SARS-CoV-2 challenge.** Although there is no attenuated SARS-CoV-2 available as a live vaccine, live viral vector COVID-19 vaccines were rapidly developed on several platforms. In addition to adenovirus vectors, paramyxovirus vectors have been used such as measles virus[16] and NDV[2,3,17]. Live vaccines typically would have an advantage over inactivated vaccines at inducing strong local innate immune responses, T-cell responses, and sterilizing mucosal antibody responses—especially, when administered mucosally. To reiterate that the design of the NDV-HXP-S renders versatility of the construct to be used as both inactivated and live vaccine, we evaluated NDV-HXP-S as a live vector vaccine in two preclinical animal models testing two different vaccination regimens. First, we examined two immunizations of live NDV-HXP-S via the IN route in the hamster model. NDV-HXP-S at a dose of $10^6$ fifty percentage of egg embryo infectious dose ($EID_{50}$) was administered to hamsters intranasally twice at day 0 and day 22. A vector-only control group was immunized with the same dose of WT NDV. A negative control group was mock-vaccinated with PBS. A healthy control group was kept unvaccinated (Fig. 5a). Serum IgG titer showed that one immunization was sufficient to induce potent binding antibody responses, whereas the booster vaccination did not further increase titers (Fig. 5b).

We speculated that the booster vaccination might generate more mucosal immunity such as spike-specific IgA, which was not measured in this study. The neutralizing activity of post-boost sera was measured against the (WT) USA-WA1/2020, and the B.1.351 and B.1.1.7 variants. A similar trend was observed as in previous studies, where immune sera neutralized the (WT) USA-WA1/2020 ($ID_{50} = 2735$) and the B.1.1.7 variant ($ID_{50} = 1819$) equally well and crossed-neutralized the B.1.351 variant ($ID_{50} = 341$) with a reduced potency (Fig. 5c). Upon challenge with $10^5$ PFU of the USA-WA1/2020 strain, we observed that the NDV-HXP-S-vaccinated animals show no weight changes like the healthy control group, whereas the negative control group lost a substantial amount of weight by day 5. Animals vaccinated with the WT NDV exhibited weight loss that was less pronounced than that of the negative control group. This could possibly be due to the innate antiviral response induced by the live WT NDV (Fig. 5d). On day 2 post infection, viruses were cleared in the lungs of animals that received the NDV-HXP-S, whereas both WT NDV and PBS groups showed similar high viral titers in the lung homogenates. In the nasal washes of WT NDV and PBS control animals, viral titers are comparably high, while only one animal out of three in the NDV-HXP-S group showed measurable viral titer (Fig. 5e). In summary, this study confirmed the effectiveness of the NDV-HXP-S as a live viral vector vaccine when administered intranasally. The vaccine not only prevented SARS-CoV-2 replication in the lungs but also significantly reduced virus shedding from the nasal cavity, which would greatly diminish the risk of virus transmission as well.

In addition to the hamster study testing only the IN route of the live vaccine, we performed a mouse study evaluating a different immunization regimen of the live NDV-HXP-S, combining the IN and intramuscular routes aiming to bring both mucosal and systemic immunity into action. Of note, the live nature of the NDV-HXP-S would have an adjuvant effect compared with an inactivated NDV-HXP-S. Here we examined this vaccination strategy with three different doses, in which the animals received the same titer of live NDV-HXP-S for the IN prime and the intramuscular boost. Three groups of animals were immunized with $10^4$ $EID_{50}$, $10^5$ $EID_{50}$, and $10^6$ $EID_{50}$ of NDV-HXP-S, respectively. A vector-only control group was immunized with $10^6$ $EID_{50}$ of the WT NDV. The negative control group was mock-vaccinated with PBS. The two immunizations were 3 weeks apart. Mice were again sensitized with Ad5-hACE2 as described earlier and challenged with $10^5$ PFU of the USA-WA1/2020 strain (Fig. 6a). By ELISA we observed a dose-dependent antibody titer, in which the high-dose group developed the strongest antibody responses after each immunization (Fig. 6b). IgG subclasses ELISAs showed a favorable induction of IgG2a over IgG1 in all NDV-HXP-S vaccine groups (Supplementary Fig. 1b). As expected, mice vaccinated with the high-dose developed the highest level of neutralizing/cross-NAbs to the WT and variant SARS-CoV-2 (Fig. 6c). In terms of protection, a reverse correlation of the vaccine dose and viral load in the lung homogenates was observed (Fig. 6d); the high-dose regimen conferred the best protection among the three groups. A duplicate experiment including the high-dose group ($10^6$ $EID_{50}$ of NDV-HXP-S) and vector-only control group ($10^6$ $EID_{50}$ of WT NDV) was set up to measure spike-specific IgA in the nasal washes, which were collected 21 days after IN prime. A spike-specific IgA ELISA showed that animals that received the NDV-HXP-S developed IgA on their respiratory mucosal surfaces after the IN prime. The IgA titer was further enhanced after the intramuscular boost (Supplementary Fig. 3). However, by intracellular cytokine staining (ICS), we did not detect more T cells responses in mice that were vaccinated with $10^6$ $EID_{50}$ of NDV-HXP-S than those in mice that were vaccinated with $10^6$ $EID_{50}$ of WT NDV at

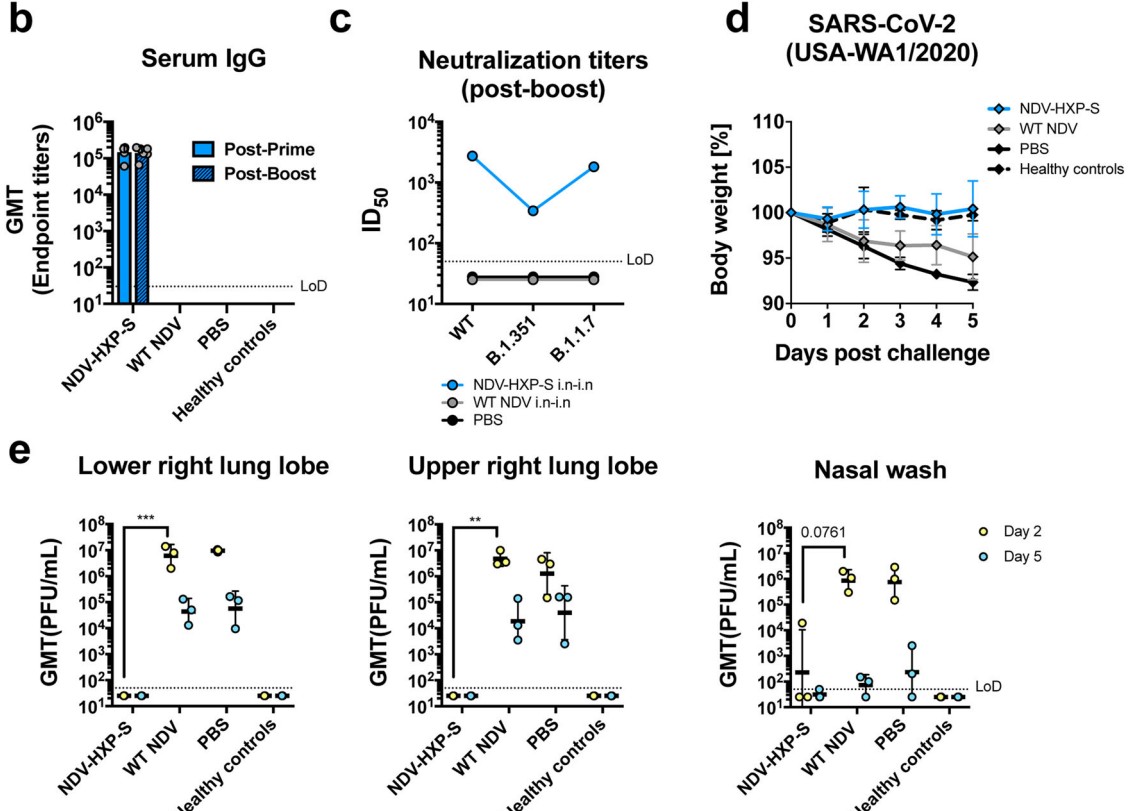

**Fig. 5 Live NDV-HXP-S via the intranasal route induces protective antibody responses in hamsters. a** Design of the study. Eighteen-to-twenty-week-old female Golden Syrian hamsters were used. Group 1 ($n = 6$) was vaccinated with $10^6$ $EID_{50}$ of live NDV-HXP-S. Group 2 ($n = 6$) was vaccinated with $10^6$ $EID_{50}$ of live WT NDV as the vector-only control. Group 3 ($n = 6$) was vaccinated with PBS as the negative control. Group 4 ($n = 6$) were the healthy controls. The vaccine was administered via the intranasal (I.N.) route at D0 and D22. Blood was collected at D22 and D41. Group 1–3 were challenged with $10^5$ PFU of the USA-WA1/2020 strain at D44. Group 4 was mock-challenged with PBS. **b** Spike-specific serum IgG. Antibodies in post-prime (D22, solid blue bar) and post-boost (D41, pattern blue bar) sera were measured by ELISAs. GMT endpoint titers were graphed. The error bars represent geometric SD. **c** Neutralizing activity of serum antibodies. Post-boost sera from groups 1–3 were pooled within each group and tested in neutralization assay against USA-WA/2020 strain (WT), B.1.351 variant, and B 1.1.7 variant in technical duplicate. Serum dilutions inhibiting 50% of the infection ($ID_{50}$) were plotted (LoD = 1:50; An $ID_{50}$ = 1:25 was assigned to negative samples). **d** Body weight change of hamsters. Bodyweight was recorded for 5 days after the challenge. The error bars represent geometric SD. (**c** and **d**, blue: NDV-HXP-S; gray: WT NDV; black: PBS) **e** Viral load in the lungs and nasal washes. The lower right and upper right lung lobes of a subset of animals ($n = 3$) from each group were collected at day 2 (yellow) and day 5 (blue) post challenge. Each lung lobe was homogenized in 1 mL PBS. Nasal washes were collected in 0.4 mL of PBS. Viral titers were measured by plaque assay on Vero E6 cells and plotted as GMT of PFU/mL (LoD = 50 PFU/mL; a titer of 25 PFU/mL was assigned to negative samples). The error bars represent geometric SD. Statistical difference was analyzed by two-way ANOVA corrected for Dunnett's multiple comparisons test (**$p < 0.0017$; ***$p = 0.0009$).

3 weeks after the boost upon stimulation with spike-specific peptides of the splenocytes (Supplementary Fig. 5).

**NDV-HXP-S is effective against SARS-CoV-2 variants of concern in the mouse model.** With the emergence of variants of concern that are partially resistant to NABs raised against the WT SARS-CoV-2 attributed to the amino-acid substitutions or deletions in the N-terminal domain (NTD) and the receptor-binding domain, we performed vaccination in mice with inactivated

NDV-HXP-S and challenged them with USA-WA1/2020, hCoV-19/USA/MD-HP01542/2021 JHU (B.1.351) and hCoV-19/Japan/TY7-503/2021 (P.1) after Ad5-hACE2 treatment. This was to ensure efficient replication of all three viruses, although both B.1.351 and P.1 variants have been reported to extend their host tropism to murine ACE2[18]. The vaccination group received an intramuscular injection of 1 µg of inactivated NDV-HXP-S, whereas the negative control group received 1 µg of the inactivated WT NDV. Two doses were administered, 3 weeks apart.

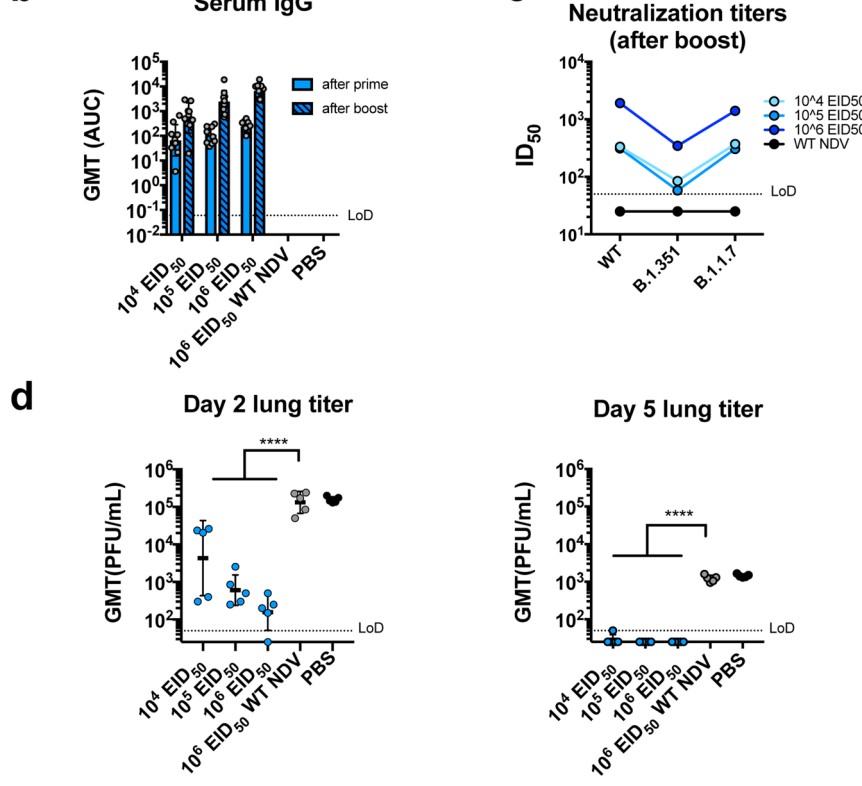

**Fig. 6 Intranasal prime followed by an intramuscular boost of live NDV-HXP-S induces protective antibody responses in mice. a** Design of the study. Seven-to-nine-week-old female BALB/c mice were used. Group 1–3 (*n* = 10) were vaccinated with $10^4$, $10^5$, and $10^6$ $EID_{50}$ of live NDV-HXP-S, respectively. Group 4 (*n* = 10) was vaccinated with $10^6$ $EID_{50}$ of WT NDV. Group 5 (*n* = 10) was mock-vaccinated with PBS. The vaccine was administered via the intranasal (I.N.) route at D0 and intramuscular (I.M.) route at D21. Blood was collected at D21 and D43. Mice were sensitized with Ad5-hACE2 at D40 and challenged with $10^5$ PFU of the USA-WA1/2020 strain at D45. **b** Spike-specific serum IgG. Antibodies in post-prime (D21, solid blue bars) and post-boost (D43, pattern blue bars) sera were measured by ELISAs. GMT AUC was graphed. The error bars represent geometric SD. **c** Neutralizing activity of serum antibodies. Post-boost sera from groups 1–4 were pooled within each group and tested in neutralization assay against USA-WA/2020 strain (WT), B.1.351 variant, and B 1.1.7 variant in technical duplicate (cyan: $10^4$ $EID_{50}$; light blue: $10^5$ $EID_{50}$; blue: $10^6$ $EID_{50}$; black: WT NDV). Serum dilutions inhibiting 50% of the infection ($ID_{50}$) were plotted (LoD = 1:50; An $ID_{50}$ = 1:25 was assigned to negative samples). **d** Viral load in the lungs. Lungs of a subset of animals (*n* = 5) from each group were collected at day 2 and day 5 post challenge (blue: NDV-HXP-S; gray: WT NDV; black: PBS). The whole lungs were homogenized in 1 mL PBS. Viral titers were measured by plaque assay on Vero E6 cells and plotted as GMT of PFU/mL (LoD = 50 PFU/mL; a titer of 25 PFU/mL was assigned to negative samples). The error bars represent geometric SD. Statistical difference was analyzed by ordinary one-way ANOVA corrected for Dunnett's multiple comparisons test (****$p$ < 0.0001).

One-third of the mice from each group were challenged with USA-WA1/2020, hCoV-19/USA/MD-HP01542/2021 JHU (B.1.351), and hCoV-19/Japan/TY7-503/2021 (P.1), respectively. Lungs of animals from each group were harvested on day 2. Viral loads were measured as described earlier. The NDV-HXP-S reproducibly inhibited WT virus replication to a great magnitude, while it also reduced B.1.351 replication by a factor of ~1000 and P.1 replication by a factor of ~280 at day 2 post challenge (Fig. 7). This study demonstrated that with expected reduction to neutralize the variants of concern, NDV-HXP-S is still effective at robustly inhibiting virus replication in vivo, which would be essential to mitigate disease.

## Discussion

The COVID-19 pandemic has promoted the unprecedented development of various vaccine platforms from the conventional inactivated whole-virion vaccines to novel mRNA vaccines, recombinant protein subunit vaccines, and viral vector vaccines[19]. The high efficacy of some novel vaccines is not necessarily followed by universal accessibility, due to logistical barriers (e.g. cost, manufacturing infrastructure, supply of raw materials, production capacity, transportation/storage, and distribution). NDV was identified as an avian pathogen in the 1920's and was later developed as an oncolytic agent and veterinary vaccine using non-virulent strains[20–25]. Similar to many other

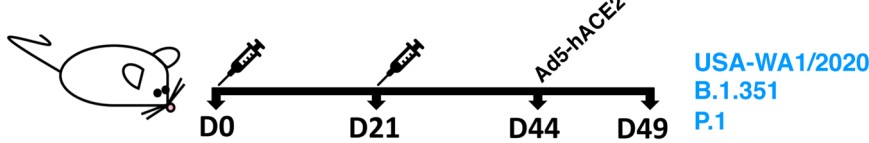

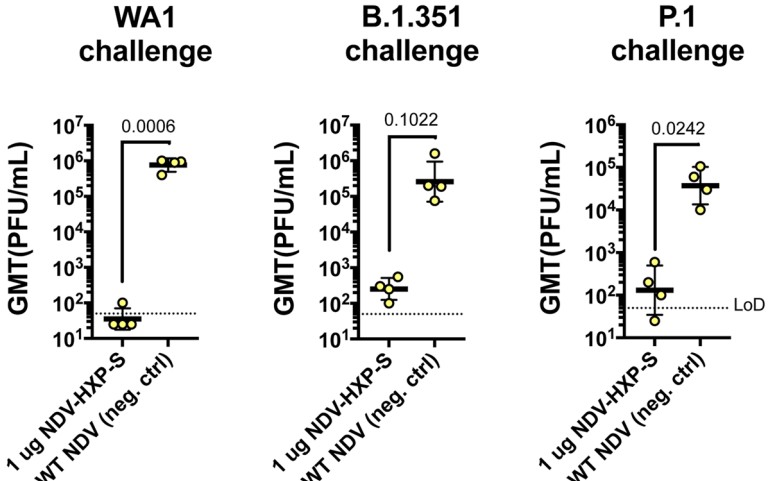

**Fig. 7 Inactivated NDV-HXP-S induces protective antibody response against the challenge of SARS-CoV-2 variants of concern.** Eight-to-ten-week-old female BALB/c mice were either vaccinated with 1 μg of inactivated NDV-HXP-S (n = 4) or WT NDV (n = 4, negative control). Two immunizations were performed via the intramuscular route at D0 and D21. At D44, mice were treated with Ad5-hACE2. At D49, one-third of mice from each group was challenged with USA-WA1/2020, B.1.351, or P.1 strain. On day 2, lungs were harvested and homogenized in 1 mL PBS. Viral titers were measured by plaque assay on Vero E6 cells and plotted as GMT of PFU/mL (LoD=50 PFU/mL; A titer of 25 PFU/mL was assigned to negative samples). The error bars represent geometric SD. Statistical difference was analyzed by a one-tailed *t* test. The *p* values are indicated.

paramyxoviruses, NDV tolerates large insertions and has been evaluated as a vaccine vector for a number of animal pathogens in preclinical studies[22,25–33]. To increase the surface expression of the inserted gene, the transmembrane domain and cytoplasmic tail of the target membrane protein can be replaced by that of HN or F protein of NDV[32]. Using this strategy, we have previously constructed a viral vector (NDV expressing the S-F chimera) and shown that it can be used as an inactivated vaccine against SARS-CoV-2[2]. This construct was later optimized to further stabilize the S-F by introducing Hexa Pro mutations identified via structural biology[34]. This second-generation NDV-HXP-S was quickly developed and is currently being evaluated in Phase I clinical trials in Mexico (NCT04871737, live vaccine), Vietnam (NCT04830800, inactivated vaccine), and Thailand (NCT04764422, inactivated vaccine).

Although the clinical trials are ongoing, we demonstrate here the versatility of the NDV-HXP-S in preclinical studies using mice and hamsters, stressing its effectiveness as a live vaccine or an inactivated vaccine. The live nature of the vaccine allowed for mucosal administration as well as intramuscular administration. As an avian pathogen, live NDV does not efficiently replicate in mammals due to host restriction. Local infection at the administration sites with the live NDV-HXP-S vaccine could occur due to the ubiquitous expression of sialic acid (the receptor of NDV). This would help to stimulate innate antiviral responses, which possibly confer protection and shape the memory immune responses differently from those associated with the inactivated vaccine[35]. In addition, vaccination with live NDV would potentially induce more cellular immunity than the inactivated vaccines. However, a pilot experiment examining the T-cell responses in the spleens by ICS (after spike peptides stimulation)

3 weeks after the boost in mice, that were primed intranasally and boosted intramuscularly with live NDV-HXP-S, did not show evidence of T-cell responses. In future studies, other time points and pulmonary cellular immunity will be evaluated. There is also an important manufacturing consideration differentiating the live NDV vaccine from the inactivated vaccine technology. Live vaccine for humans must be propagated in specific pathogen-free (SPF) embryonated eggs, whereas inactivated vaccines for humans can be safely manufactured in non-SPF eggs, as the inactivating process eliminates the risk of avian adventitious agents arising from the use of widely available breeder chicken eggs that are one-tenth the cost and not limited in supply. A dose-ranging study in mice indicated the inactivated NDV-HXP-S was highly immunogenic even at a dose as low as 0.01 μg of the total protein (the amount of the spike protein would be much less). CpG 1018 is an approved human adjuvant that is included in the HEPLISAV-B® vaccine. Therefore, it was predominantly assessed in the present experiments with the inactivated NDV-HXP-S. The adjuvant CpG 1018 leads to an antigen-sparing effect, which is slightly better in mice than that in hamsters. It was postulated that the NDV could be self-adjuvant due to viral components (RNA and proteins) being present, resulting in a less beneficial effect of the adjuvant. However, we did observe that CpG 1018 drove a more prominent IgG2a production over IgG1 than the unadjuvanted vaccine in mice, suggesting it could promote a favorable $T_H1$ response, which will be measured in the Phase I trials that include CpG 1018 adjuvanted groups. Nevertheless, the final verdict for the usefulness of CpG 1018 in combination with NDV-HXP-S will come from the human data. Moreover, it appeared that an oil-in-water nano-emulsion animal adjuvant (AddaVax) substantially enhanced antibody titers in hamsters.

We also observed a good adjuvant effect of AddaVax in our previous studies in mice[2]. Therefore, it will be of great interest to evaluate other human adjuvants in our future preclinical studies.

It was observed that mice and hamsters vaccinated with NDV-HXP-S developed strong antibody responses that not only neutralized the prototype SARS-CoV-2 but also cross-neutralized variants of interest/concern. The reduction of neutralizing activity against B.1.351 and B.1.1.7 is consistent with what was observed for other vaccines using the prototype spike as the immunogen[6–9]. Interestingly, cross-reactivity was improved by using CpG 1018 as an adjuvant. Although NAbs correlate well with protection[36], non-NAbs could also exert protection via Fc-mediated effector functions such as antibody-dependent cellular cytotoxicity and antibody-dependent cellular phagocytosis[37]. It was also known that in mice, IgG2a subclass has a higher affinity of binding to activation FcγR than IgG1 subclass[38]. Given that CpG 1018 showed a more significant bias towards IgG2a production over IgG1 in mice, it is possible that non-neutralizing/protective antibodies were induced. However, mouse, hamster, and human TLR9 receptors are different and the activity of CpG 1018 in the ongoing clinical trials would be more relevant to the future use of this adjuvant with inactivated NDV-HXP-S. Of note, challenge studies in Ad5-hACE2 sensitized mice using B.1.351 and P.1 variants demonstrated good protection by the prototype antigen expressed by the NDV. Last but not least, this NDV platform could be quickly adapted to express the spike protein of SARS-CoV-2 variants. So far, we have successfully generated NDV-HXP-S (B.1.351[Beta]), NDV-HXP-S (B.1.1.7 [Alpha]), NDV-HXP-S (P.1 [Gamma]), and NDV-HXP-S (B.1.617.2 [Delta]) and are evaluating them in animal models. Heterologous vaccination regimens or multi-valent formulations might be beneficial to the induction of cross-protective antibodies.

## Methods

**Cells**. BSRT7 cells were a kind gift from Dr. Benhur Lee at Icahn School of Medicine at Mount Sinai (ISMMS)[39,40] and Vero E6 cells were purchased from ATCC (CRL-1586). Chicken embryo fibroblasts (CEF) were isolated as described in a previous study[41]. All cell lines were maintained in Dulbecco's Modified Eagle's Medium (DMEM; Gibco) containing 10% (vol/vol) fetal bovine serum (FBS), 100 unit/mL of penicillin, 100 µg/mL of streptomycin (P/S; Gibco) and 10 mM 4-(2-hydroxyethyl)-1-piperazineethanesulfonic acid (HEPES) at 37 °C with 5% $CO_2$.

**Plasmids**. Hexa Pro (HXP) mutations including F817P, A892P, A899P, A942P, K986P, and V987P have been identified to stabilize the prefusion conformation of spike[4] were introduced into the S-F chimera by PCR (HXP-S)[2]. The sequence of the HXP-S was inserted into pNDV_LS/L289A rescue plasmid (between P and M genes) by in-Fusion cloning (Clontech). The recombination product was transformed into MAX Efficiency™ Stbl2™ Competent Cells (Thermo Fisher Scientific) to generate the pNDV-HXP-S rescue plasmid. The plasmid was purified using PureLink™ HiPure Plasmid Maxiprep Kit (Thermo Fisher Scientific). The primers used to introduce the Hexa Pro mutations are listed in Supplementary Table 1.

**Rescue of the NDV-HXP-S**. As described in our previous studies[2], BSRT7 cells stably expressing the T7 polymerase were seeded onto six-well plates at $3 \times 10^5$ cell per well in duplicate. The next day, cells were transfected with 2 µg of pNDV-HXP-S, 1 µg of pTM1-NP, 0.5 µg of pTM1-P, 0.5 µg of pTM1-L and 1 µg of pCI-T7opt were resuspended in 250 µL of Opti-MEM (Gibco). The plasmid cocktail was then gently mixed with 15 µL of TransIT LT1 transfection reagent (Mirus). The growth media were replaced with opti-MEM during transfection. To increase rescue efficiency, BSRT7-CEF co-culture was established the next day as described previously[42]. Specifically, transfected BSRT7 cells and CEF wells were washed with warm PBS and trypsinized. Trypsinized cells were neutralized with an excessive amount of growth media. Mix BSRT7 cells with CEF cells (~1: 2.5) in a 10-cm dish. The co-culture was incubated at 37 °C overnight. The next day, the media were removed and cells were gently washed with warm PBS, opti-MEM supplemented with 1% P/S and 0.1 µg/mL of tosyl phenylalanyl chloromethyl ketone-treated trypsin was added. The co-cultures were incubated for 2 or 3 days before inoculation into 8- or 9-day old embryonated chicken eggs. To inoculate eggs, cells and supernatants were harvested and homogenized by several syringe strokes. One or two hundred microliters of the mixture were injected into each egg. Eggs were incubated at 37 °C for 3 days and cooled at 4 °C overnight. Allantoic fluid was harvested from cooled eggs and the rescue of the viruses was determined by hemagglutination (HA) assays.

**Preparation of inactivated concentrated virus**. The viruses in the allantoic fluid were first inactivated using 0.05% BPL as described previously[2]. To concentrate the viruses, allantoic fluids were clarified by centrifugation at $3441 \times g$ at 4 °C for 30 min using a Sorvall Legend RT Plus Refrigerated Benchtop Centrifuge (Thermo Fisher Scientific). Clarified allantoic fluids were laid on top of a 20% sucrose cushion in NTE buffer (100 mM NaCl, 10 mM Tris-HCl, 1 mM EDTA, pH 7.4). Ultracentrifugation in a Beckman L7-65 ultracentrifuge at $112,499 \times g$ for 2 hours at 4 °C using a Beckman SW28 rotor (Beckman Coulter) was performed to pellet the viruses through the sucrose cushion while soluble egg proteins were removed. The virus pellets were resuspended in PBS (pH 7.4). The total protein content was determined using the bicinchoninic acid assay (Thermo Fisher Scientific).

**SDS–PAGE**. The concentrated NDV-HXP-S or WT NDV was mixed with Novex™ Tris-Glycine SDS Sample Buffer (2×) (Thermo Fisher Scientific), NuPAGE™ Sample Reducing Agent (10×) (Thermo Fisher Scientific), and PBS at appropriate amounts to reach a total protein content of 20 µg in 50 µl volume. The mixture was heated at 90 °C for 5 min. The samples were mixed by pipetting and loaded at 30 µg to a 4–20% 10-well Mini-PROTEAN TGX™ precast gel (Bio-Rad). Ten microliters of the Novex™ Sharp Pre-stained Protein standard (Thermo Fisher Scientific) were used as the ladder. The electrophoresis was run in Tris/Glycine SDS/Buffer (Bio-Rad). The gel was then washed with distilled water at room temperature several times until the dye front in the gel was no longer visible. The gel was stained with 20 mL of SimplyBlue™ SafeStain (Thermo Fisher Scientific) for a minimal of 1 h to overnight. The SimplyBlue™ SafeStain was decanted and the gel was washed with distilled water several times until the background was clear. Gels were imaged using the Bio-Rad Universal Hood Ii Molecular imager (Bio-Rad) and processed by Image Lab Software (Bio-Rad).

**Sucrose-gradient fractionation**. To generate sucrose-gradient, 12.5 mL of 10% (w/v) sucrose in 0.0125 M citrate buffer was laid on top of 12.5 mL 60% (w/v) sucrose in 0.0125 M citrate buffer in round-bottom polypropylene copolymer centrifuge tubes (Thermo Fisher Scientific). The vertical tubes were then sealed using parafilm and gently flipped to a horizontal position to allow diffusion to occur at RT for 1:30 h. The tubes were then returned to a vertical position and kept in the fridge at 4 °C overnight to ensure sufficient diffusion. Five hundred microliters of the concentrated virus obtained from sucrose-cushion purification described above were laid on top of the gradient and ultracentrifugation was performed at 25,000 rpm for 2 h at 4 °C in a Beckman L7-65 ultracentrifuge. One mL of each fraction was collected with a total of 26 fractions (the last fraction is <1 mL). Twenty microliters of each fraction were mixed with 25 µl of Novex™ Tris-Glycine SDS Sample Buffer (2×) (Thermo Fisher Scientific) and 5 µl of NuPAGE™ Sample Reducing Agent (10×) (Thermo Fisher Scientific) to reach a total volume of 50 µl. The mixture was treated at 90 °C for 5 min. Thirty microliters of each sample were resolved on 4–20% of SDS–PAGE. The electrophoresis and the protein staining were performed as described above.

**Virus titration by $EID_{50}$ assays**. Fifty percent of egg $EID_{50}$ assay was performed in 9–11-day old chicken embryonated eggs. The virus in the allantoic fluid was 10-fold serially diluted in PBS, resulting in $10^{-5}$ to $10^{-10}$ dilutions of the virus. One hundred microliters of each dilution were injected into each egg for a total of 5–10 eggs per dilution. The eggs were incubated at 37 °C for 3 days and then cooled at 4 °C overnight. Allantoic fluids were collected and analyzed by HA assay. The $EID_{50}$ titer of the NDV, determined by the number of HA-positive and HA-negative eggs in each dilution was calculated using the Reed and Muench method.

**Animal experiments**. All the animal experiments were performed in accordance with protocols approved by the Icahn School of Medicine at Mount Sinai Institutional Animal Care and Use Committee (IACUC). All the animals were housing in a temperature/humidity-controlled facility with 12 h light/dark cycle. All experiments with live SARS-CoV-2 were performed in the Centers for Disease Control and Prevention/US Department of Agriculture-approved biosafety level 3 (BSL-3) biocontainment facility of the Global Health and Emerging Pathogens Institute at the Icahn School of Medicine at Mount Sinai, in accordance with institutional biosafety requirements.

**Mouse immunization and challenge studies**. Female BALB/c mice were used in all studies. For intramuscular vaccination using the inactivated NDV-HXP-S, vaccine or negative control WT NDV was prepared in 100 µl total volume with or without CpG 1018 as the adjuvant. Two immunizations were performed for all the mice at a 21-day interval. To administer live NDV-HXP-S, mice were anesthetized with ketamine/xylazine cocktail and vaccinated with NDV-HXP-S, WT NDV, or PBS in 30 µl total volume via the IN route and boosted with the same preparation via the IM route with a 21-day interval. For SARS-CoV-2 infection, mice were intranasally infected with $2.5 \times 10^8$ PFU of Ad5-hACE2 5 days prior to being challenged with $10^5$ PFU of the USA-WA1/2020 strain, $3.4 \times 10^4$ PFU of the

hCoV-19/USA/MD-HP01542/2021 JHU strain (B.1.351, kindly provided by Dr. Andrew Pekosz from Johns Hopkins Bloomberg School of Public Health) or $6.3 \times 10^4$ PFU of the hCoV-19/Japan/TY7-503/2021 strain (P.1). Viral titers in the lung homogenates of mice 2 days or 5 days post infection were used as the readout for protection. In brief, the lung lobes were harvested from a subset of animals per group and homogenized in 1 mL of sterile PBS. Viral titers in the lung homogenates were measured by plaque assay on Vero E6 cells. To collect nasal washes, mice were euthanized and nasal washes were collected in 1 mL PBS containing 0.1% bovine serum albumin (BSA), 10 units/mL penicillin, and 10 µg/mL streptomycin. The nasal washes were spun at 3000 rpm for 20 min at 4 °C and stored at −80 °C. Blood was collected by submandibular vein bleeding. Sera were isolated by low-speed centrifugation and stored at −80 °C until use.

**Hamster immunization and challenge studies.** Female Golden Syrian hamsters were used in all the studies. For intramuscular vaccination study, NDV-HXP-S vaccine or negative control WT NDV/PBS was prepared in 100 µL of total volume either without adjuvants or with 100 µg of CpG 1018 or 50 µL of AddaVax as the adjuvant. For the IN vaccination study, hamsters were anesthetized with ketamine/xylazine cocktail before the IN administration of live NDV-HXP-S, WT NDV, or PBS in a 50 µL volume. An unvaccinated healthy control group was included in each study. The animals were vaccinated following a prime-boost regimen in a ~3-week interval. In all, 2–3 weeks after the boost, animals were challenged with $10^4$ or $10^5$ PFU of the USA-WA1/2020 strain, except that the healthy control group was mock-challenged with the same amount of PBS. Animals were bled via a lateral saphenous vein and sera were isolated by low-speed centrifugation. Weight changes of the animals were monitored for 5 days. A subset of animals from each group was euthanized at day 2 and day 5 post challenge to harvest lungs lobes (upper right lung lobe, lower right lung lobe), nasal turbinates, or nasal washes. Each right lung lobe was homogenized in 1 mL of PBS. The nasal turbinates were homogenized in 0.5 mL of PBS. Nasal washes were collected in 0.4 mL PBS. Viral titers in the nasal washes, nasal turbinate, and lung homogenates were measured by plaque assay on Vero E6 cells. The left lung lobes were collected at day 5 and fixed/perfused with neutral buffered formalin for histopathology. Of note, the challenge study using GMP vaccine from the three collaborating manufacturers was conducted using similar methods for assessments of outcomes, but with vaccine formulations from each manufacturer containing 1 µg of S protein per dose, with or without CpG 1018.

**ELISAs.** ELISAs were performed as described previously[2,3] to measure spike-specific IgG in the serum of mice and hamsters vaccinated with NDV-HXP-S. In brief, Immulon 4 HBX 96-well ELISA plates (Thermo Fisher Scientific) were coated with 2 µg/mL of recombinant trimeric S protein (50 µL per well) in 1× coating buffer diluted from 10× coating buffer (SeraCare Life Sciences Inc.) overnight at 4 °C. The plates were then washed three times with 220 µL PBS containing 0.1% (v/v) Tween-20 (PBST). The plates were subsequently blocked with 220 µL blocking solution (3% goat serum, 0.5% dry milk, 96.5% PBST) per well for 1 h at RT. The blocking buffer was decanted and mouse or hamster sera were threefold serially diluted in blocking solution starting at 1:30 dilution followed by a 2 h incubation at RT. The plates were washed three times with 220 µL PBST and 50 µL of Amersham ECL sheep anti-mouse IgG HPR-linked whole secondary antibody (Cytiva, NA931) or HRP-conjugated goat anti-hamster IgG (H+L) cross-adsorbed secondary antibody (Invitrogen, HA6007) was added at 1:3000 dilution. The plates were incubated for 1 h at RT and washed 3 times with 220 µL PBST. One hundred µL of o-phenylenediamine dihydrochloride (SigmaFast OPD, Sigma) substrate was added per well. The plates were developed for 10 min and 50 µL of 3 M hydrochloric acid (HCl) was added to each well to quench the reactions. The optical density (OD) was measured at 492 nm on a Synergy 4 plate reader (BioTek) or similar. An average of OD values for blank wells plus three standard deviations was used to set a cutoff value for each plate. The area under the curve (AUC) or endpoint titers were calculated and graphed using GraphPad Prism 7.0e. Similarly, to measure spike-specific IgG1 or IgG2a subclass in the mouse sera, an HRP-conjugated goat anti-mouse IgG1 (Abcam, ab97240) or an HRP-conjugated goat anti-mouse IgG2a (Abcam, ab97245) were used at 1:3000 dilution. To measure spike-specific IgA in the nasal wash, the clarified undiluted nasal washes were used

and serially diluted by 2-fold. An HRP-conjugated goat anti-mouse IgA secondary antibody (Bethyl laboratories, #90-103 P) was used at 1:2000 dilution.

**Microneutralization assays using the authentic SARS-CoV-2 viruses.** The microneutralization assays using the authentic SARS-CoV-2 viruses were described previously[43]. In brief, serum samples were heat-inactivated at 56 °C for 60 min prior to use. In all, 2× minimal essential medium (MEM) supplemented with glutamine, sodium biocarbonate, HEPES, and antibiotics P/S was used for the assay. Vero E6 cells were maintained in culture using DMEM supplemented with 10% FBS. Twenty-thousand cells per well were seeded the night before in a 96-well cell culture plate. In all, 1× MEM was prepared from 2× MEM and supplemented with 2% FBS. Threefold serial dilutions of pooled serum starting at 1:20 or 1:50 dilution were prepared in a 96-well cell culture plate and each dilution in 80 µL was mixed with 600 times the 50% tissue culture infectious dose ($TCID_{50}$) of SARS-CoV-2 in 80 µL. The Serum-virus mixture was incubated for 1 h at room temperature and then was added to the cells for 1 h and kept in a 37 °C incubator. Next, the virus-serum mixture was removed and the corresponding serum dilution was added to the cells with addition 1× MEM. The cells were incubated for 2 days and fixed with 100 µL of 10% formaldehyde per well for 24 h before being taken out of the BSL-3 facility. The staining of the cells was performed in a biosafety cabinet (BSL-2). The formaldehyde was removed from the cells. Cells were washed with 200 µL PBS once before being permeabilized with PBS containing 0.1% Triton X-100 for 15 min at RT. Cells were washed with PBS again and blocked in PBS containing 3% dry milk for 1 h at RT. Cells were then stained with 100 µL per well of a mouse monoclonal anti-NP antibody (1C7C7), kindly provided by Dr. Thomas Moran at ISMMS, at 1 µg/ml for 1 h at RT. Cells were washed with PBS and incubated with 100 µL per well Anti-mouse IgG HRP (Rockland) secondary antibody at 1:3000 dilution in PBS containing 1% dry milk for 1 h at RT. Finally, cells were washed twice with PBS and the plates were developed using 100 µL of SigmaFast OPD substrate. Ten minutes later, the reactions were stopped using 50 µL per well of 3 M HCl. The OD 492 nM was measured on a Biotek SynergyH1 Microplate Reader or similar. Non-linear regression curve fit analysis (The top and bottom constraints are set at 100% and 0%) over the dilution curve was performed to calculate 50% of inhibitory dilution ($ID_{50}$) of the serum using GraphPad Prism 7.0e. To compare the neutralization titers of immune sera from animals vaccinated with NDV-HXP-S, the neutralization assays against the wild type SARS-CoV-2 (isolate USA-WA1/2020), hCoV-19/South Africa/KRISP-K005325/2020 (B.1.351, BEI Resources NR-54009), and hCoV-19/England/204820464/2020 (B.1.1.7, BEI Resources NR-54000) were performed at the same time to avoid assay-to-assay variations.

**Pseudo-particle neutralization assays.** The PNA was performed by Nexelis using a replication-incompetent vesicular stomatitis virus, displaying the spike protein of Wuhan-Hu-1 strain[15].

**SARS-CoV-2 plaque assay.** The plaque assay was performed in BSL-3 facility. Vero E6 cells were seeded onto 12-well plates in growth media at 1:5 and were cultured for 2 days. Tissue homogenates or nasal washes were 10-fold serially diluted in the infection medium (DMEM containing 2% FBS, P/S, and 10 mM HEPES). Two hundred microliters of each dilution were inoculated onto each well starting at 1:10 dilution of the sample. The plates were incubated at 37 °C for 1 h with occasional rocking every 10 min. The inoculum in each well was then removed and 1 mL of agar overlay containing 0.7% of agar in 2× MEM was placed onto each well. Once the agar was solidified, the plates were incubated at 37 °C with 5% $CO_2$. Two days later, the plates were fixed with 5% formaldehyde in PBS overnight before being taken out of BSL-3 for subsequent staining in BSL-2 cabinet. The plaques were immuno-stained with an anti-SARS-CoV-2 NP primary mouse monoclonal antibody 1C7C7 kindly provided by Dr. Thomas Moran at ISMMS. An horseradish peroxidase (HRP)-conjugated goat anti-mouse secondary antibody was used at 1:2000 and the plaques were visualized using TrueBlue[TM] Peroxidase Substrate (SeraCare Life Sciences Inc.).

**Table 1 Scoring system used to define the pathological changes in the lungs.**

| Score | Area affected | Epithelial degeneration/necrosis | Inflammation |
|---|---|---|---|
| 0 | None | None | None |
| 1 | 5–10% | Minimal; scattered cell necrosis/vacuolation affecting 5 to 10% of tissue section | Minimal; scattered inflammatory cells affecting 5–10% of tissue section |
| 2 | 10–25% | Mild; scattered cell necrosis/vacuolation | Multifocal, few inflammatory cells |
| 3 | 25–50% | Moderate; multifocal vacuolation or sloughed/necrotic cells | Thin layer of cells (<5 cell layer thick) |
| 4 | 50–75% | Marked; multifocal/segmental necrosis, epithelial loss/effacement | Thick layer of cells (>5 cell layer thick) |
| 5 | >75% | Severe; coalescing areas of necrosis, parenchymal effacement | Confluent areas of inflammation |

**Histopathology and immunohistochemistry**. Formalin-fixed, paraffin-embedded left lung tissues obtained from hamsters were cut into 5 μm sections and stained with hematoxylin and eosin (H&E) by the Biorepository and Pathology Core. All sections were evaluated by a veterinary pathologist who was blinded to the vaccination groups in the Comparative Pathology Laboratory (CPL) at ISMMS. The scoring system used was described in Table 1 below. For immunohistochemistry, the sections were counterstained using hematoxylin, and the nucleoprotein (N) of SARS-CoV-2 was stained using a mouse anti-N monoclonal antibody 1C7C7 (1:50 dilution) kindly provided by Dr. Thomas Moran at ISMMS. The slides were scanned using OptraSCAN Scanner.

**Spleen processing and ICS**. Spleens were homogenized manually through a 40-μm cell strainer, and red blood cells were lysed by the addition of distilled water for 10 seconds. Splenocytes were resuspended in complete RPMI media (cRPMI, Gibco, Thermo Fisher Scientific, Waltham, MA, USA) supplemented with 10% w/v FBS (Gibco), 100 U/mL of Penicillin-Streptomycin (Gibco), and 2mM L-Glutamine (Gibco). Cells were seeded in V-bottom 96-well plates (CELLSTAR, Greiner Bio-One North America Inc., Monroe, NC, USA) at an average of $2 \times 10^6$ cells/well in cRPMI media containing anti-mouse CD28 (1:500, BD Biosciences, Franklin Lakes, NJ, USA), brefeldin A (1:1,000, GolgiPlug™, BD Biosciences), and monensin (1:1,000, GolgiStop™, BD Biosciences). Splenocytes were stimulated with PepMix™ SARS-CoV-2, a pool of 158 + 157 peptides (15-mer peptides overlapping by 11 aa) spanning the full length of Spike Glycoprotein (PM-WCPV-2, JPT Peptides, Berlin, Germany), at a final individual peptide concentration of 5 μg/mL at 37 °C with 5% CO2 for 6 h. Negative control cells were stimulated with an equivalent volume of DMSO. Positive control cells were stimulated with a cocktail containing phorbol 12-myristate 13-acetate (0.5 mg/mL, Sigma-Aldrich, St. Louis, MO, USA) and ionomycin (1 mg/mL, Sigma-Aldrich, St. Louis, MO, USA). The unstimulated control cells were only treated with the complete RPMI media. After stimulation, cells were washed with PBS containing 2% FBS and centrifuged at 1200 rpm for 5 min and then stained with Zombie Red™ diluted in PBS (1:500, BioLegend, San Diego, CA, USA) for 15 min at RT in the dark. Cells were washed in PBS containing 2% FBS (1200 rpm for 5 min) and incubated with surface staining cocktail containing Fc Block CD16/CD32 1:50 (BD Biosciences) and the anti-mouse antibodies BV 711 CD3 (1:400), Pacific Blue CD4 (1:200), PerCP/Cy5.5 CD8 (1:200) (BioLegend) for 30 min at 4 °C in fluorescence-activated cell sorting (FACS) buffer (PBS containing 0.1% BSA and 2 mM EDTA). Cells were washed in FACS buffer and then incubated in fixation/permeabilization buffer (BD Biosciences) for 5 min at 4 °C. After fixation, cells were washed in 1× permeabilization buffer (BD Biosciences), then incubated with the intracellular staining cocktail containing anti-mouse antibodies Alexa Fluor 647 IFN-γ (1:200), Alexa Fluor 488 TNF-α (1:100), PE/Cy7 IL-2 (1:100) in 1× permeabilization buffer for 30 min at 4 °C. Samples were then washed in 1× permeabilization buffer and resuspended in PBS buffer for acquisition. Samples were acquired on an Aurora spectral cytometer (Cytek, Fremont, CA, USA) using SpectroFlo® software (Cytek), with the relevant single fluorochrome compensation controls set by the daily acquisition of Cytometer Setup and Tracking beads. Analysis was performed with FCS Express 7 (DeNovo Software) and GraphPad Prism 8.0.2 (GraphPad Software).

**Statistical analysis**. The statistical analysis was performed using GraphPad Prism 7.0e or 8.0.2. For multiple comparison, the statistical difference was determined using ordinary one-way ANOVA or two-way ANOVA with Dunnett's correction. To compare the two groups, a one-tailed $t$ test was used.

**Reporting summary**. Further information on research design is available in the Nature Research Reporting Summary linked to this article.

## Data availability
Source data are provided with this paper.

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

## Acknowledgements

We thank Dr. Benhur Lee to kindly share the BSRT7 cells. We thank Dr. Thomas Moran for the 1C7C7 antibody. We thank Dr. Robert Coffman and Dynavax for providing the CpG 1018. We thank Dr. Andrew Pekosz from Johns Hopkins Bloomberg School of Public Health for providing the B.1.351 challenge virus. We thank Dr. Randy Albrecht for support with the BSL-3 facility, procedures and management of import/export at the Icahn School of Medicine at Mount Sinai, New York. We also thank Nexelis for performing the PNA. This work was partially supported by an NIAID-funded Center of Excellence for Influenza Research and Surveillance (CEIRS, HHSN272201400008C, P.P.) and a grant from an anonymous philanthropist to Mount Sinai (P.P., F.K., A.G.-S.). This work was also supported, in part, by the Bill & Melinda Gates Foundation [INV-021239]. Under the grant conditions of the foundation, a Creative Commons Attribution 4.0 generic License has already been assigned to the Author Accepted Manuscript version that might arise from this submission. The findings and conclusions contained within are those of the authors and do not necessarily reflect the positions or policies of the Bill & Melinda Gates Foundation. K.S. is supported by the Japanese Society for the Promotion of Science (JSPS) Overseas Research Fellowship.

## Author contributions

Conceptualization and design: P.P., F.K., AG-S., and W.S.; construction, preparation and characterization of the vaccines: W.S., S.M., Y.L., S.S., I.G., V.R., N.L., J.M.; GMP lots vaccine preparation: IVAC (DH.T.), GPO (P.W.), Instituto Butantan (R.O.), PATH (B.L.I., R.S., R.H., R.R.); mouse immunization and in vitro serological assays: W.S., Y.L., I.G., K.S.; mouse and hamster challenge and plaque assays: W.S., Y.L., I.G.; micro-neutralization assay: F.A.; data analysis: P.P., W.S., Y.L., I.D., F.A., A.G-S., F.K.; virus reagents: F.A., L.C., M.S., I.M., R.R.A. (ISMMS), S.J.; first draft of the manuscript: P.P. and W.S.; manuscript review and editing, all authors.

## Competing interests

The Icahn School of Medicine at Mount Sinai has filed patent applications entitled "RECOMBINANT NEWCASTLE DISEASE VIRUS EXPRESSING SARS-COV-2 SPIKE PROTEIN AND USES THEREOF" which names P.P., F.K., AG-S., and W.S. as inventors. The AG-S laboratory has received research support from Pfizer, Senhwa Biosciences, Kenall Manufacturing, Avimex, Johnson & Johnson, Dynavax, 7Hills Pharma, Pharmamar, ImmunityBio, Accurius, Merck and Nanocomposix, and AG-S has consulting agreements for the following companies involving cash and/or stock: Vivaldi Biosciences, Contrafect, 7Hills Pharma, Avimex, Vaxalto, Pagoda, Accurius, Esperovax, Farmak, Applied Biological Laboratories, and Pfizer. All other authors declared no competing interests.
