## [Peer Review File · Nature Communications]

A Newcastle disease virus expressing a stabilized spike protein of SARS-CoV-2 induces protective immune responsesReviewers' Comments:

Reviewer #1:

Remarks to the Author:

Sun et al.

The manuscript describes preclinical studies of a SARS-CoV-2 vaccine based on a Newcastle Disease Virus platform, as both, an inactivated or a live vaccine. These vaccines are currently in clinical trials. The platform takes advantage of existent and wide-spread technologies for virus production in chicken eggs, that could increase its production around the world at a reduce cost. The manuscript shows convincingly that the vaccine, in the inactivated form given intramuscularly or the live vaccine given intranasally, protects mice and hamsters in vaccination, boosting, and challenge studies. The vaccines generate total and neutralizing antibodies at similar levels as seen in human convalescent serum samples. Antibodies induced by the vaccines neutralized all three SARS-CoV-2 variants tested () although at a lesser extent the X and the X variants, as expected. However, challenge studies with all three variants in the mouse model show that vaccine confers strong lung protection, despite being less than that conferred against the prototype variant. Overall, the manuscript well written, with very detailed and simple figures, and with convincing protection results that are consistent between two tested animal models, demonstrating that this is a vaccine with potential to become an efficacious alternative for vaccination against SARS-CoV-2. The manuscript if of high quality and will generate strong interest.

Minor points:

Figure 3 Table. Change the route of immunization to NA for the HC instead of IM.

Figure 3B: The effect of adjuvants is not clear as indicated in lines 165-166. Statistics?

Line 231, since IgA was measured, move Fig. 5B presentation one sentence up where IgG is described.

Line 235, how the neutralizing titer by NDV-HXP-S in Fig. 5B compares with the inactivated one (e.g., Fig. 3C).

Lane 271 should read "substitution"

Reviewer #2:

Remarks to the Author:

In this study, Sun et al. reported the generation of an NDV-vectored covid-19 vaccine carrying a prefusion-stabilized version of S protein (NDV-HXP-S). They provided the results of a series of pre-clinical assessments, including the immunogenicity and the protective efficacy in two animal models: Ad5-hACE2 treated mice and Golden Syrian Hamsters. They showed the feasibility of large-scale production for this candidate. They also tested the possibility to use NDV-HXP-S as a live vectored vaccine. By challenging the vaccinated animals with different SARS-CoV-2 isolates, the authors demonstrated that this candidate could confer effective protection against variants of concern. Generally, the results are interesting and facilitate the development of an effective vaccine for the developing countries. However, several issues must be addressed before this paper can be published.

Major

1. In the majority of challenge assays, the authors did not provide histopathology results. An effective vaccine should not only inhibit viral replication and shedding, but should suppress disease progression. Thus, histopathology results, especially those from mice experiments, should be provided.
2. Regarding the immunization assay with live NDV-HXP-S, T cell response should be assessed. If the live NDV-HXP-S can infect mouse or hamster cells, the S protein can be expressed in vivo, and S-specific T cell response (especially CD8+T cells) should be elicited. This may be another advantage of a live vectored vaccine.
3. Regarding the intranasal immunization with live NDV-HXP-S, more details of the local immunity should be examined and provided, including the pulmonary T cell response and the neutralizing

activity of the bronchoalveolar lavage fluids. These data may further strengthen the major conclusions of this work.

4. Which kind of immune response determines the protective efficacy of NDV-HXP-S? In Fig. 3C and D, the NT titer of NDV-HXP-S alone group was higher than the CpG 1018 group, but the protection was not higher accordingly; Fig. 4C and D, there is no significant difference about the NT titers among different immunization groups, but why the IVAC+CpG group did not show higher protection than the IVAC alone group as other candidates did?

Minor

1. Fig.1B, a western-blotting and other assays should be performed to confirm the presence of S protein on viral surface.

2. It seemed that AddaVax functioned very well in the hamster model, but why was this adjuvant not assessed in mice model?

3. Line 168, the CpG adjuvant seemed to improve the quality of neutralizing antibody response, why? Can the authors give some discussions about this phenomenon?

4. Line 189, here why the AddaVax adjuvant was not assessed in parallel?

5. Fig.4 and Fig.S2, the viral loads in the lung tissues were effectively controlled throughout the experiment, and no significant inflammation was observed, but why the challenged animals that received vaccination showed a transient loss of the body weight? In my opinion, additional histopathology analysis should be performed at day 2 or 3 after challenge.

6. A brief introduction about NDV vector should be added. Regarding the live version, the tissue tropism and the infectivity of this vector may affect the outcome of the immunogenicity.

Point-by-point response to the reviewers' comments

REVIEWER COMMENTS

Reviewer #1 (Remarks to the Author):

Sun et al.

The manuscript describes preclinical studies of a SARS-CoV-2 vaccine based on a Newcastle Disease Virus platform, as both, an inactivated or a live vaccine. These vaccines are currently in clinical trials. The platform takes advantage of existent and wide-spread technologies for virus production in chicken eggs, that could increase its production around the world at a reduce cost. The manuscript shows convincingly that the vaccine, in the inactivated form given intramuscularly or the live vaccine given intranasally, protects mice and hamsters in vaccination, boosting, and challenge studies. The vaccines generate total and neutralizing antibodies at similar levels as seen in human convalescent serum samples. Antibodies induced by the vaccines neutralized all three SARS-CoV-2 variants tested () although at a lesser extent the X and the X variants, as expected. However, challenge studies with all three variants in the mouse model show that vaccine confers strong lung protection, despite being less than that conferred against the prototype variant. Overall, the manuscript well written, with very detailed and simple figures, and with convincing protection results that are consistent between two tested animal models, demonstrating that this is a vaccine with potential to become an efficacious alternative for vaccination against SARS-CoV-2. The manuscript if of high quality and will generate strong interest.

Minor points:

Figure 3 Table. Change the route of immunization to NA for the HC instead of IM.

Response: We changed “IM” to “NA” as suggested (Figure 3a)

Figure 3B: The effect of adjuvants is not clear as indicated in lines 165-166. Statistics?

Response: We have performed statistical analysis on Figure 3b as suggested.

Line 231, since IgA was measured, move Fig. 5B presentation one sentence up where IgG is described.

Response: “Fig. 5b” was moved one sentence up (Line 237)

Line 235, how the neutralizing titer by NDV-HXP-S in Fig. 5B compares with the inactivated one (e.g., Fig. 3C).

Response: We now provide ID₅₀ values in the text for Figure 5b and 3c for direct comparison of the neutralization titers across different experiments (Lines 169-171; Lines 241-242). The neutralization titer from live vaccine immunization (WT: ID₅₀ = 2735; B.1.351: ID₅₀ = 341; B.1.1.7: ID₅₀=1819) in Fig. 5b is similar to those in the unadjuvanted inactivated vaccine group (WT: ID₅₀ = 2429; B.1.351: ID₅₀ = 425; B.1.1.7=2710) in Figure 3c

Lane 271 should read “substitution”

Response: Thank you for pointing out the typo. It’s corrected (Line 283)

Reviewer #2 (Remarks to the Author):

In this study, Sun et al. reported the generation of an NDV-vectored covid-19 vaccine carrying a prefusion-stabilized version of S protein (NDV-HXP-S). They provided the results of a series of pre-clinical assessments, including the immunogenicity and the protective efficacy in two animal models: Ad5-hACE2 treated mice and Golden Syrian Hamsters. They showed the feasibility of large-scale production for this candidate. They also tested the possibility to use NDV-HXP-S as a live vectored vaccine. By challenging the vaccinated animals with different SARS-CoV-2 isolates, the authors demonstrated that this candidate could confer effective protection against variants of concern. Generally, the results are interesting and facilitate the development of an effective vaccine for the developing countries. However, several issues must be addressed before this paper can be published.

Major

1. In the majority of challenge assays, the authors did not provide histopathology results. An effective vaccine should not only inhibit viral replication and shedding, but should suppress disease progression. Thus, histopathology results, especially those from mice experiments, should be provided.

Response: We agree with the reviewer that pathological changes in the lungs are essential indicators of SARS-CoV-2 induced disease progression. Mice are not naturally susceptible to wildtype SARS-CoV-2 infection. Transduction of Ad5-hACE2 might result in non-specific inflammation, although such impact is minimal 5 days after Ad5-hACE2 treatment (Rathnashinghe et al., Emerging Microbes & Infections). With the hamster model also available, we thought it would be better to examine lung histopathology in the hamsters, in which SARS-CoV-2 infection could naturally induce lung pathological changes. We now provided additional representative H&E staining and immunohistochemistry (IHC) images for hamster lung histopathology as Supplementary Fig. 2f. A corresponding description was added (Lines 214-215, 542-545, 913-918)

2. Regarding the immunization assay with live NDV-HXP-S, T cell response should be assessed. If the live NDV-HXP-S can infect mouse or hamster cells, the S protein can be expressed in vivo, and S-specific T cell response (especially CD8+T cells) should be elicited. This may be another advantage of a live vectored vaccine.

Response: We agree with the reviewer. We had actually performed an intracellular staining experiment to examine T cells responses in the spleens as shown in Supplementary Figure 5 (i.n prime-i.m boost in mice with live NDV-HXP-S). We did not

observe an increase of spike-specific T cell responses in mice that were vaccinated with the NDV-HXP-S as compared to those in mice that were vaccinated with the WT NDV (vector-only) controls at 3 weeks after the boost upon stimulation with spike peptides. We have added information related to this data to the manuscript (Results: Lines 276-279; Material and methods: Lines 567-596; Figure legend: Lines 932-937) and a discussion to that effect (Lines 324-329)

3. Regarding the intranasal immunization with live NDV-HXP-S, more details of the local immunity should be examined and provided, including the pulmonary T cell response and the neutralizing activity of the bronchialveolar lavage fluids. These data may further strengthen the major conclusions of this work.

Response: The reviewer made a good point that for an intranasally administered live vaccine, local immune responses should be characterized. We have obtained more data on S-specific IgA induced locally (in the nasal wash) after the boost pertaining to Figure 6 and S3 (i.n. prime-i.m. boost in mice). We have now revised Figure S3 to include this data. We also added a description of this result in the text (Lines 275-276, 923-924)

4. Which kind of immune response determines the protective efficacy of NDV-HXP-S? In Fig. 3C and D, the NT titer of NDV-HXP-S alone group was higher than the CpG 1018 group, but the protection was not higher accordingly; Fig. 4C and D, there is no significant difference about the NT titers among different immunization groups, but why the IVAC+CpG group did not show higher protection than the IVAC alone group as other candidates did?

Response: Neutralization titer typically correlates well with protection in preclinical animal models and humans (Khoury et al., Nature Medicine, 2021). But non-neutralizing antibodies could also execute protection *in vivo*, such as those that activate antibody-dependent cellular cytotoxicity (ADCC) and antibody-dependent cellular phagocytosis (ADCP). We added a discussion with respect to this point. The following sentences have been added “Although neutralizing antibodies correlate well with protection, non-neutralizing antibodies could also exert protection via Fc-mediated effector functions such as antibody-dependent cellular cytotoxicity (ADCC) and antibody-dependent cellular phagocytosis (ADCP). It was also known that in mice, IgG2a subclass has higher affinity of binding to activation Fc γ R than IgG1 subclass. Given that CpG 1018 showed a more significant bias towards IgG2a production over IgG1 in mice, it is possible that non-neutralizing/protective antibodies were induced. However, mouse, hamster and human TLR9 receptors are different and the activity of CpG 1018 in the ongoing clinical trials would be more relevant to the future use of this adjuvant with inactivated NDV-HXP-S.” (Lines 353-361).

Regarding the lack of antibody enhancement of CpG 1018 with IVAC vaccine, it might be a result of slightly different manufacturing methods. We added a statement to line 220-221

Minor

1. Fig.1B, a western-blotting and other assays should be performed to confirm the presence of S protein on viral surface.

Response: In a sucrose gradient of purified virus, we observed that the spike protein co-migrated with NDV viral proteins, demonstrating that the spike is indeed incorporated into NDV virions. This data is now shown as Supplementary Figure 4. Corresponding changes were also made in the text (Lines 105-112, 431-444, 926-930)

2. It seemed that AddaVax functioned very well in the hamster model, but why was this adjuvant not assessed in mice model?

Response: AddaVax is not approved for use in humans. In the present mouse study, we would like to thoroughly assess CpG 1018, because it was later used in Phase I clinical trials in Thailand and Vietnam. This has been added to discussion (Lines 336-338, 345-347)

3. Line 168, the CpG adjuvant seemed to improve the quality of neutralizing antibody response, why? Can the authors give some discussions about this phenomenon?

Response: Please see major point 4. We have added discussions (Lines 350-358).

4. Line 189, here why the AddaVax adjuvant was not assessed in parallel?

Response: Please see minor point 2 for discussion added to Lines 336-338, 345-347.

5. Fig.4 and Fig.S2, the viral loads in the lung tissues were effectively controlled throughout the experiment, and no significant inflammation was observed, but why the challenged animals that received vaccination showed a transient loss of the body weight? In my opinion, additional histopathology analysis should be performed at day 2 or 3 after challenge.

Response: We observed the transient loss of body weight up to day 2 or 3 after challenge in vaccinated animals, which is probably the result of low amount of virus replication in the upper respiratory tracts. We have added a sentence (Line 209)

6. A brief introduction about NDV vector should be added. Regarding the live version, the tissue tropism and the infectivity of this vector may affect the outcome of the immunogenicity.

Response: Thank you for the suggestion, we have added a brief statement in the discussion. The following sentences have been added “As an avian pathogen, NDV does not efficiently replicate in mammals due to host-restriction. A local infection at the administration sites with the live NDV-HXP-S vaccine could occur due to the ubiquitous expression of sialic acid (the receptor of NDV). This would help to stimulate innate antiviral responses, which possibly confer protection and shape the memory immune responses differently from those associated with the inactivated vaccine” (Line 320-324)

Reviewers' Comments:

Reviewer #2:

Remarks to the Author:

The authors addressed most my comments. I have no further questions.